# Mosquito Blood Feeding Prevention Using an Extra-Low DC Voltage Charged Cloth

**DOI:** 10.3390/insects14050405

**Published:** 2023-04-23

**Authors:** Kun Luan, Marian G. McCord, Andre J. West, Grayson Cave, Nicholas V. Travanty, Charles S. Apperson, R. Michael Roe

**Affiliations:** 1Department of Textile Engineering, Chemistry and Science, Wilson College of Textiles, NC State University, Raleigh, NC 27695, USA; kluan@ncsu.edu; 2Department of Forest Biomaterials, College of Natural Resources, NC State University, Raleigh, NC 27695, USA; marianmccord@gmail.com; 3Department of Textile and Apparel, Technology and Management, Wilson College of Textiles, NC State University, Raleigh, NC 27695, USA; ajwest2@ncsu.edu; 4Department of Entomology & Plant Pathology, College of Agriculture and Life Sciences, NC State University, Raleigh, NC 27695, USA; glcave@ncsu.edu (G.C.); travanty@gmail.com (N.V.T.); apperson@ncsu.edu (C.S.A.)

**Keywords:** mosquito, electric, textile, probing, protection, low-voltage

## Abstract

**Simple Summary:**

An extra-low, DC (direct current) voltage field applied to a textile was used to prevent mosquito blood feeding. A novel 3-D textile was developed based on the mosquito head structure that when charged with 15 volts was 100% effective in preventing mosquito blood feeding across an artificial membrane.

**Abstract:**

Mosquito vector-borne diseases such as malaria and dengue pose a major threat to human health. Personal protection from mosquito blood feeding is mostly by treating clothing with insecticides and the use of repellents on clothing and skin. Here, we developed a low-voltage, mosquito-resistant cloth (MRC) that blocked all blood feeding across the textile and was flexible and breathable. The design was based on mosquito head and proboscis morphometrics, the development of a novel 3-D textile with the outer conductive layers insulated from each other with an inner, non-conductive woven mesh, and the use of a DC (direct current; extra-low-voltage) resistor-capacitor. Blockage of blood feeding was measured using host-seeking *Aedes aegypti* adult female mosquitoes and whether they could blood feed across the MRC and an artificial membrane. Mosquito blood feeding decreased as voltage increased from 0 to 15 volts. Blood feeding inhibition was 97.8% at 10 volts and 100% inhibition at 15 volts, demonstrating proof of concept. Current flow is minimal since conductance only occurs when the mosquito proboscis simultaneously touches the outside layers of the MRC and is then quickly repelled. Our results demonstrated for the first time the use of a biomimetic, mosquito-repelling technology to prevent blood feeding using extra-low energy consumption.

## 1. Introduction

Blood feeding by arthropods such as mosquitoes, ticks, lice, and flies is a major threat to human health causing diseases such as malaria, dengue fever, and Lyme disease [1]. These arthropods carry microbial pathogens that are transmitted to humans when a blood meal is acquired [2]. Personal protection from vector-important arthropods typically involves the use of chemical treatments, i.e., insecticide (permethrin)-treated fabrics and the application of repellents such as DEET (*N,N*-diethyl-*m*-toluamide) to exposed skin and clothing; these can have deleterious effects if not properly used [3,4,5]. Regardless of whether repellents and insecticides are considered by regulatory agencies as safe, many people are reluctant to use them. Pyrethroid-treated clothing is becoming ineffective because of mosquito resistance to the insecticide, and there are concerns about human exposure to these pesticides [6]. Females who are pregnant or trying to become pregnant and children are especially at risk to vector-borne disease [7,8,9]. Therefore, there is an urgent need to improve the current insecticide-based protection systems or to develop an alternative, effective technology to prevent blood feeding, and which could eliminate or at least reduce the use of chemical treatments.

Mosquito blood feeding on human skin is a combination of complex insect–skin mechanical [10] and chemical interactions [11]. Mosquitoes have developed body, head, and mouthpart morphometrics and specialized behaviors to blood feed in seconds without detection, which is essential for their survival [12]. Blood feeding involves proboscis probing and sewing and insertion into the skin [12]. Blocking these movements could deter or prevent feeding [13], the focus of this paper.

It is well known that electrical charges can affect insect behavior [14] and have been correlated with insect avoidance [15]. However, avoidance typically requires a high voltage field, as shown in several studies [16,17,18], which is not practical for garments. Other electrical devices utilize an electrified screen where mosquitoes and other insects are drawn into contact with the screen by light and chemical attractants [15]. These technologies use a high voltage circuit or voltage booster to kill or repel mosquitoes and other insects [14,19]. The working voltage in these devices exceeds what is safe for humans (~36V DC) causing discomfort and damage if contacted. It would be challenging and dangerous to incorporate this application into clothing.

From a study of the physics of textile–mosquito interactions, we developed mathematical models to describe textile structure that would prevent mosquito blood feeding, including their use in garments [13]. These models now serve as a “mathematical road map” for producing textiles without insecticides that physically block blood feeding. We have advanced these studies [13,20,21] in this paper to develop an electrical, mosquito-resistant cloth (MRC) charged with an extra-low, DC (direct current) working voltage that blocks blood feeding. An in vitro feeding bioassay system was developed to test the protective capacity of the MRC against mosquito blood feeding. The capability for incorporating the MRC into garments is discussed. Our work to incorporate a static electric field into textiles demonstrates proof of concept for a “new way” of achieving mosquito avoidance without applying chemicals to clothing.

## 2. Materials and Methods

### 2.1. Mosquitoes

In this study, 7–8 d old female adult yellow fever mosquitoes, *Aedes aegypti*, were used to investigate mosquito blood feeding on our electrical mosquito-resistant cloth. The mosquitoes were reared at 27 ± 1 °C, 80% relative humidity, and with a 14:10 h light:dark cycle by methods describe before [13].

*Aedes aegypti* head morphometrics was used to design the physical structure of the MRC (structure and principle of function for the MRC are described in more detail later). Mosquito morphometrics (head diameter and proboscis length and diameter) were measured with a digital microscope (Monozoom-7 Zoom Microscope, Bausch and Lomb, Bridgewater, NJ, USA) and a Phenom G1 desktop scanning electron microscope (SEM; Thermo Fisher Scientific Inc., Waltham, MA, USA) in the Phenom SEM and Forensic Textile Microscopy Laboratory at North Carolina State University, Raleigh, NC, USA.

### 2.2. In Vitro Feeding/Bioassay System

Figure 1 shows the in vitro feeding bioassay system for testing the electrified cloth (MRC). The feeding system includes a Plexiglas^®^ cage (Weiterstadt, Germany) that contains the mosquitoes, a blood reservoir (Figure 1C), and a circulating water bath to regulate the temperature (~37 °C) of the blood (Figure 1B) [13]. The operational window on the Plexiglas^®^ cage (Figure 1A,B) was used to insert the blood reservoir into the cage and served as a conduit for the stainless-steel tubes (and circulating warm water) and for the power cables to the electrified cloth (described in more detail later). During testing, the blood reservoir was covered with a collagen film (product code 894010.95; Devro, Inc., Columbia, SC, USA) with the electrified cloth on top of the collagen film (Figure 1C). The membrane and blood serve as artificial human skin, and the heat attracts mosquitoes to the surface. They insert their proboscis across the cloth and collagen membrane to blood feed. Each test used 100 host-seeking, first ovipositional cycle, unfed *Aedes aegypti* females. Blood was from the cow, *Bos Taurus*, and the blood was purchased from Chaudhry Hala Meats, Siler City, NC, USA, treated with sodium citrate (0.2%, wt/vol) to prevent clotting, transported to our laboratory at 4 °C, and stored in small aliquots at −80 °C in our laboratory until needed. Blood was never re-frozen.

### 2.3. Design of the Mosquito-Resistant Cloth (MRC)

Mosquito blood feeding on human skin is the result of a series of behaviors allowing the mouth parts to be inserted into blood vessels close to the skin surface. When the mosquito lands on skin and then probes, mouthpart chemoreceptors interact with the host skin to identify a blood-rich location. Then needle-like structures pierce the epithelium and are inserted into a blood vessel, where the exchange of saliva and blood occurs. The strategy in the design of the MRC was to terminate this process early enough to prevent blood feeding. Using the collagen membrane, a worst-case scenario for preventing blood feeding was established, since the artificial membrane is much thinner and easier to penetrate than the cornified squamous epithelium of human skin, and mosquitoes did not have to search for a blood vessel as is needed for human skin.

For the MRC, we designed and constructed a sandwich structure stacked with two layers of conductive fabrics (conductive fabric-1 and conductive fabric-2) insulated from each other by a middle layer of fiberglass mesh fabric (Figure 2A; more details on construction methods for the MRC described later). Each pore on the fabric is similar to a “switch” that can turn the resistor-capacitor (RC) circuit on and off (Figure 2A and B, respectively). When the proboscis touches the outer conductive layers, the current supplied by an external power source travels through the proboscis (Figure 2B). The hypothesis is the conductance of current through the proboscis at a low voltage will prevent blood feeding through the collagen membrane using the in vitro bioassay system (Figure 1). Since the mosquito is standing on the top surface of the textile, an alternative mechanism of action is conductance through the legs, thorax, head, and proboscis when the latter makes contact with the bottom layer of the MRC and where the proboscis is not touching the top layer. This is theoretically possible but less likely because of the pore size of the top conductive layer relative to the diameter of the proboscis, the irregular shape of the pores (described more later), the erratic probing behavior of the mosquito through the MRC, and the increased resistance that would occur when current has to flow through the legs, thorax, head, and proboscis.

The sandwich structure of the MRC that was tested (Figure 3A) consists of two layers of conductive knits on the top and bottom insulated electrically from each other by a single layer of fiberglass woven mesh in the middle. The top and bottom layers were glued to the middle layer of the fiberglass woven mesh with a 3M 77_TM_ Multipurpose Adhesive (3M, St. Paul, MN, USA). Two leads were constructed for the electrical connection to the positive and negative leads on the power supply by a simple extension of one each of the outer conductive knits (Figure 3B). The conductive material was a Nylon knit coated with silver particles (product code SMS-0109; Swift Textile Metalizing LLC, Bloomfield, CT, USA). The insulation mesh was a woven textile (Figure 3C) composed of 33% fiberglass, 66% PVC, and 1% undefined other materials (Senneny, Wuxi, China). Figure 3D shows the surface pattern of the conductive fabric, a single jersey construction.

The 3-D structure of the MRC including pore size and thickness were key parameters to allow mosquito probing through the sandwiched fabric structure and connectivity of the two conductive layers (Figure 2). Pores in the knitted fabric had irregular shapes, in which the measurement of the knit fabric pore diameter was obtained only by digital microscopy and estimated using ImageJ software (1.52t 30 January 2020), as described before [13]. Fabric thickness was measured with a Thwing-Albert ProGage Thickness Tester (Thwing-Albert ProGage instrument company, West Berlin, NJ, USA), as described in ASTM D1777.

### 2.4. In Vitro Bioassay

In vitro bioassays were conducted in the mosquito insectary laboratory at the Dearstyne Entomology Building at NC State University at 27 ± 1 °C, 75–80% humidity, and during the normal photophase in which the insects were reared under florescent overhead lighting. After the blood reservoir (16.5 cm length × 3.5 cm width × 0.5 cm depth; Figure 1C) was assembled and filled using a 30 mL syringe containing animal blood, the MRC sample was placed between the gasket and collagen film. Then the blood reservoir was assembled and held in place by metal binder clips. Prior to testing, a circulating water bath was used to warm the blood to 37 °C. Then 100 *Aedes aegypti* unfed females were released into the cage for the bioassay.

A power supply (Model KD3005P; KORAD programmable DC power supply, SRA Soldering Products, Walpole, MA, USA) was used to charge the MRC with a directional DC voltage from 0 to 30 V. A multimeter (Fluke FLUKE-115 Digital Multimeter, Fluke Corporation, Everett, WA, USA) was used to verify the output voltage of the power supply. The voltage measured both by the power supply and the multimeter were in exact agreement. We first tested 30 V as a trial to examine the repellency of the MRC. Then, to further determine the minimum voltage for 100% repellency, we charged the MRC with 15, 10, 5, and 0 V. Bioassays were replicated 3 times at each voltage. The duration of each bioassay was 10 min, during which the number of times females landed on the barrier material was counted. A landing was defined as contact for at least 1 s where the mosquito stood on the MRC and regardless of whether the mosquito probed. In some cases, the same mosquito could have landed multiple times and some mosquitoes may not have landed at all. After the bioassay, mosquitoes were removed from the cage, frozen, and crushed on a sheet of white paper. For a worst-case scenario, any red color appearing on the paper for each insect was counted as an indication of blood feeding.

### 2.5. Temperature Monitoring of MRC

To eliminate any possible heating effect from the application of voltage on mosquito feeding, the surface temperature of the MRC was measured at each minute during the bioassay for 10 min at 15 volts using an infrared camera (FLIR TG165 Thermal Imager, Teledyne FLIR LLC, Billerica, MA, USA).

### 2.6. Data Analysis

The morphometrics for each parameter taken was replicated from 10 different mosquitoes from which a mean and standard deviation were calculated using ORIGINPRO^®^ 2018. Bioassay results were plotted with the same software using a box chart to summarize the statistical values. The solid brown dot in the box charts represents the raw data. The height of the box represents the 25th and 75th percentiles. The whiskers represent the 5th and 95th percentiles. Additional values included the median (line inside of the box) and mean (white dot), which are also presented in the box chart. Bioassay data were analyzed by one-sample Student’s *t*-test, where NS is not significant, ** *p* < 0.001, and *** *p* < 0.0001.

## 3. Results

### 3.1. Parameters in the Design of the MRC

The *Aedes aegypti* female morphometrics are shown in Figure 4 and were used to design and manufacture the MRC. Materials selected, when combined into the three layers of the MRC, had a thickness less than the proboscis length (Figure 4A) and pore sizes that were larger than the proboscis diameter but smaller than the head diameter (Figure 4B). In the case of the latter, the mosquito head was excluded physically from being inserted below the surface of the MRC. The method for the mosquito to take a blood meal across the MRC was to bridge the thickness of the three layers with their proboscis only. The hypothesis was that proboscis insertion across the MRC toward the blood feeding membrane would connect the two conductive layers, generating a closed circuit (Figure 2B). Current flow through the proboscis would result in cessation of blood feeding.

### 3.2. Bioassay Results

Figure 5 shows the number of mosquito landings in response to different levels of charge on the MRC. Since 0 voltage was expected to have no influence on mosquito probing and blood feeding and insects when blood feeding are not flying, this should reduce the landing count. Moreover, those fully blood fed across the MRC are no longer attracted to the feeding membrane, migrate to the walls of the plexiglass cage, and become quiescent, lowering the landing count. If feeding across the membrane when a charge was present prevents blood feeding and repels the insects from the surface of the MRC, it was expected these insects would continue to host seek and repeatedly land on the surface of the MRC, probe, and be repelled again (increasing landings). Although there was a trend of higher landing counts at 5–15 V compared to 0 V as expected, the differences were not statistically significant (5 volts, *p* = 0.06211; 10 volts, *p* = 0.09264; 15 volts, *p* = 0.10032; Student’s *t*-test).

Figure 6 shows the relationship of the number of blood fed mosquitoes versus voltage. Five volts DC reduced the mean number of blood feds, but the difference was not significant. Blood feeding was minimal at 10 V (97.8% feeding inhibition) and no blood feeding occurred at 15 V, demonstrating proof of concept for our MRC.

### 3.3. MRC Surface Temperature

We examined the temperature of the MRC at the highest voltage tested, 15 V, to assess whether temperature had any role in preventing blood feeding. No differences were found during the 10 min bioassay (average 35.4 °C; Figure 7B). The temperature on the surface of the MRC was in the normal range of adult human skin, 33–37 °C.

## 4. Discussion

The average proboscis length of *Aedes aegypti* females was 2.2 mm, which required an MRC thickness < 2.2 mm to allow the mosquitoes to penetrate the cloth and feeding membrane to obtain a blood meal. Furthermore, for exclusive proboscis penetration of the MRC, the pore size of the conductive fabrics had to be smaller than the mosquito head (810 μm) but larger than the proboscis diameter (80 μm). Therefore, the criteria for designing the MRC were as follows:(1)$$L_{probosics}<T$$
(2)$$D_{probosics}<P<D_{head}$$
where T was the MRC thickness; P was the pore size of the MRC; and $L_{probosics}$, $D_{probosics}$, and $D_{head}$ were the mosquito proboscis length, diameter, and head diameter, respectively. Any sandwiched RC circuit satisfying Equations (1) and (2) would then provide an electrical repellence for blood feeding by closing an electrical circuit between the outside charged layers of the MRC, as illustrated in part in Figure 2B. During probing, the proboscis passes through the insulating fiberglass layer and makes contact with the bottom conductive knit. Current flow can occur through the mosquito body, thorax head and proboscis (since the mosquito is standing on the top layer) or by a shorter pathway when the proboscis touches the top layer (Figure 2B). 

Figure 8A–C shows the change in electrical potential during mosquito probing through the fabric specific for the proboscis only contacting the MRC two layers. Two possible repellent principles are illustrated, where (Figure 8B) the electrical potential is transferred to the proboscis as it approaches the top and bottom conductive layers, and where current is flowing through the proboscis (Figure 8C), producing an uncomfortable feeling and/or damages the proboscis. The changes in electrical potentials in both of these conditions (Figure 8B,C), and whether current is passing through the legs, body, and proboscis since the mosquito is standing on the top of the cloth (discussed earlier), could not be measured in our proof-of-concept studies and will require, in the future, high-resolution measurements of electrical potential differences and current flow around the site of the insect–MRC interaction and between the MRC, insect body, and proboscis (using a different bioassay and circuit architecture).

The following equations describe a mathematical model for the MRC mechanism of action and power consumption for the condition illustrated in Figure 8C. Two assumptions were made: (i) the two conductive knit layers were parallel to each other, and (ii) all the electrical power accumulated on the conductive fabrics will pass through the mosquito proboscis at the moment when the two conductive layers are connected. The definition for parallel plate capacitance is:(3)$$C=\frac{Q}{V}=\varepsilon_{0}\times \frac{A}{d}$$
where *C* is the capacitance of the parallel plates (F); *Q* is the electric charge (C); *V* is the voltage (V); $\varepsilon_{0}$ is the permittivity of air (1.0006 F/m); *A* is the area of the plate (m^2^); and *d* is the separation distance (m). Thus, approximation of the charge released from the MRC (*Q*) is:(4)$$Q=\varepsilon_{0}\times \frac{A}{d}\times V$$

The current (*I*) passing through the proboscis is:(5)$$I=\frac{Q}{t}$$

Based on Ohm’s law, the conductive resistance of the insect (*R*) is:(6)$$R=\frac{V}{I}=\frac{td}{\varepsilon_{0}A}$$

The power consumption when the proboscis (*P*) completes the circuit would be:(7)$$P=I^{2}R=\frac{\varepsilon_{0}A}{td}\times V^{2}$$

Power consumption of the MRC only occurs when the mosquito bridges the two outer conductive layers of the cloth. Therefore, current flow is limited by proboscis resistance in the simplest case or by flow through the body and proboscis with the mosquito standing on the MRC and the proboscis not touching the top layer. This flow is also rapidly stopped when the mosquito is repelled from probing across the MRC. There is no power consumption when mosquitoes are not probing. Moreover, in typical mosquito exposures in a real-world situation, the number of landings and probings would be much lower than in the worst-case scenario in our bioassay with 100 mosquitoes confined to a small space around the feeding probe. The clear advantage of the MRC is operation at much lower voltages than previous approaches of applying high electrical potentials to control mosquitoes (presented in the introduction), the application of our technology to a textile suitable to make garments to prevent mosquito blood feeding (discussed later), and minimal non-target effects on human subjects because of the low energy input, i.e., low voltage and current flow and the short duration of current flow.

There are some challenges in moving this proof of concept into a practical garment to prevent mosquito blood feeding. One consideration is developing a commercial scale production method for the MRC irrespective of how it might be used in a garment. One straightforward approach would be 3-D knitting where the yarns used to make the inner layer would be non-conductive while the outside layers are constructed using conductive yarns. The use of 3-D knitting to make comfortable and durable mosquito textiles and garments that block mosquito blood feeding, based on textile structure alone, were described before by our group [13]. Our models and knitting approaches described in this application could be a starting point to develop a next-generation MRC for a particular garment use. The alternative would be lamination, as practiced in the current study. A core effort in both approaches will be the development of the conductive yarns and textiles that have the desirable flexibility, durability, and other features required for comfort as a garment and especially for a climate where mosquitoes would be active and biting. The second challenge is how to incorporate the MRC into a garment. Using “cut and sew” techniques, this would be less challenging if areas of the garments where mosquitoes prefer to blood feed, such as the back and shoulders, are incorporated with the MRC.

There are now three approaches to block blood feeding across a textile, i.e., application of chemicals (insecticides or repellents), the use of textile structure [13], and the use of our low-voltage MRC technology. There are obvious advantages and disadvantages of each approach. For example, the use of chemicals can result in human exposure to pesticides. Use of structure alone, depending on what structural choices are made (discussed in more detail in [13]), can affect comfort. The MRC eliminates the need for pesticides and in some uses could improve comfort but requires an energy source. These different approaches could be used in combination. However, the final solution will certainly change with the intended use of the textile, environmental conditions of use, economics of garment construction, and risks associated with the exposure level to disease-infected mosquitoes.

There are other considerations of functionality for the use of the MRC, such as the use of an “incidence detection circuit” to provide the user with an alarm that mosquitoes are trying to blood feed. Because of the stealth nature of mosquitoes, people are often bitten before they are even aware of exposure. Other adaptations could include the use of a cell phone to power and provide smart functions to the MRC. Smart functions could include blood-feeding monitoring and the control of power usage under a variety of environmental conditions such as precipitation. Constructing an entire garment with a fully functional MRC will be more challenging.

## 5. Conclusions

In this study, we developed a mosquito-resistant cloth that can provide 100% protection from mosquito blood feeding. Power consumption occurs only when the mosquito bridges the two outer conductive layers of the MRC with its proboscis. Power consumption is limited by proboscis and/or body resistance and the rapid cessation of mosquito probing and repellency from the MRC. An in vitro bioassay system was developed for future studies of electrically activated textiles to prevent mosquito blood feeding. We anticipate the biomimicry technology described here can lead to low-voltage, electrical, insect-repellent materials. The addition of current monitoring and incident counting could also be used as a personal detection method for clothing where real-time mosquito interactions with garments and mosquito biting pressure can be automatically monitored.

## Figures and Tables

**Figure 1 insects-14-00405-f001:**
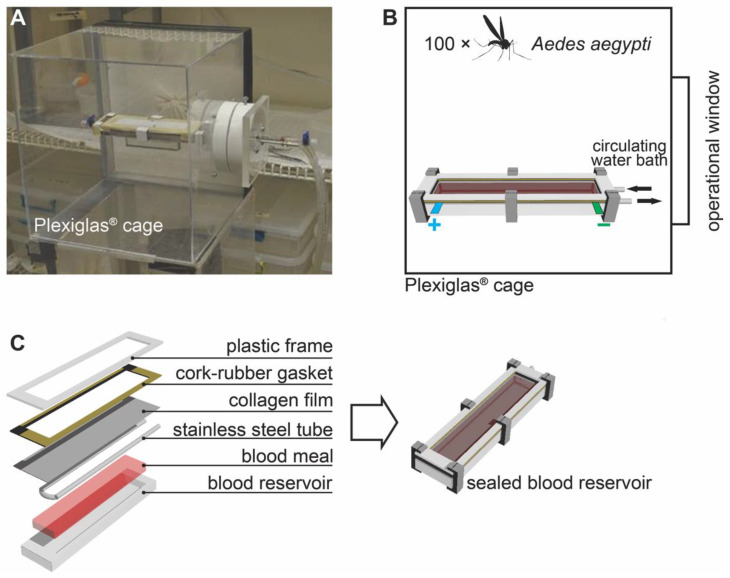
In vitro feeding/bioassay system: (**A**) setup image of in vitro feeding cage; (**B**) schematic of in vitro bioassay cage with 100 *Aedes aegypti* females for evaluating percentage blood feeding across the MRC; and (**C**) assembly of the blood reservoir (the MRC was placed between the gasket and collagen film).

**Figure 2 insects-14-00405-f002:**
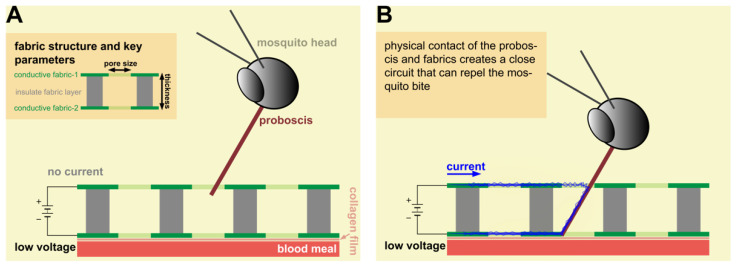
Diagram of the mosquito-resistant cloth (MRC) on the collagen membrane with the blood reservoir below (also shown in Figure 1C with MRC missing). (**A**) Insert shows the structure of the resistor-capacitor (RC) circuit of the MRC; the position of the mosquito proboscis where conductance and feeding inhibition is hypothesized not to occur; and (**B**) position of the proboscis allowing conductance and where feeding inhibition is hypothesized to be initiated (described in more detail in the insert).

**Figure 3 insects-14-00405-f003:**
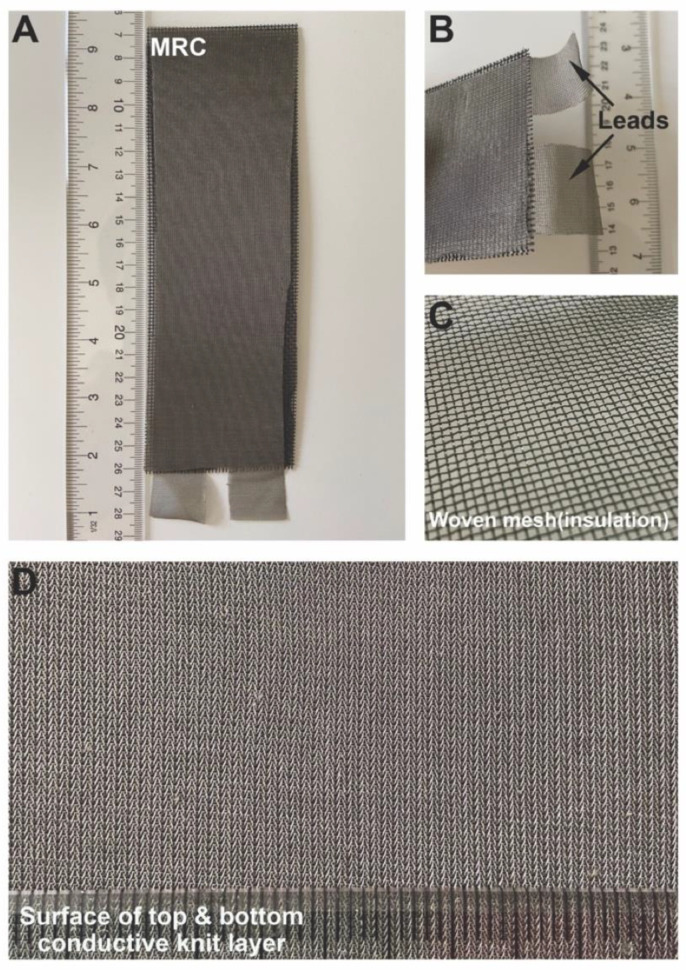
Mosquito-resistant cloth (MRC) that was tested. (**A**) photograph of MRC used for testing; (**B**) shows the electrical leads for connecting the MRC to a power supply; (**C**) insulating middle layer of fiberglass mesh; and (**D**) surface pattern of the conductive knit. See text for details for the construction of the MRC.

**Figure 4 insects-14-00405-f004:**
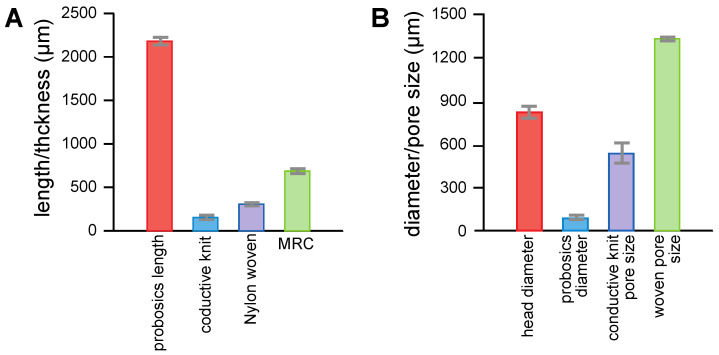
Mosquito morphometrics used to design the MRC. (**A**) Length of the proboscis versus the thickness of the MRC; and (**B**) diameter of the head and proboscis versus the pores in the MRC. The error bars are ±1 standard deviation.

**Figure 5 insects-14-00405-f005:**
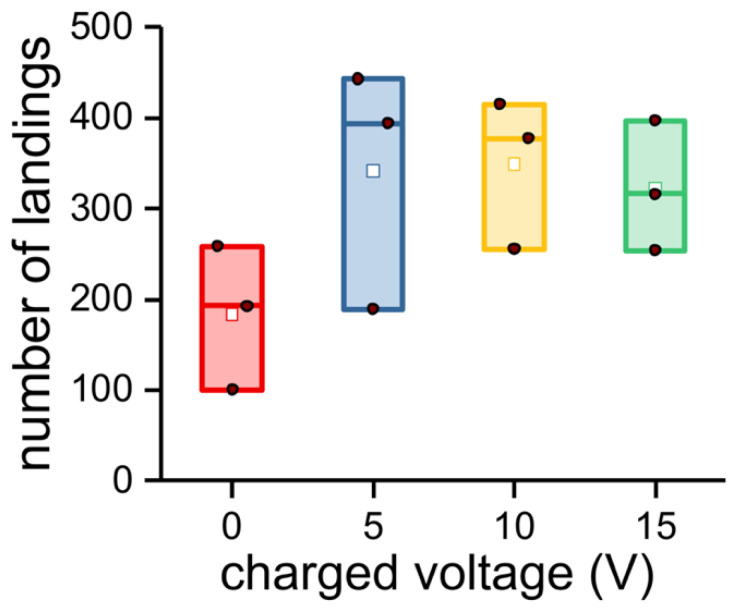
Number of landings. The solid brown dot in the box charts represents the raw data. The height of the box depends on the 25th and 75th percentiles. The whiskers represent the 5th and 95th percentiles. Additional values including the median (line inside of the box) and mean (white dot) are also presented in the box chart.

**Figure 6 insects-14-00405-f006:**
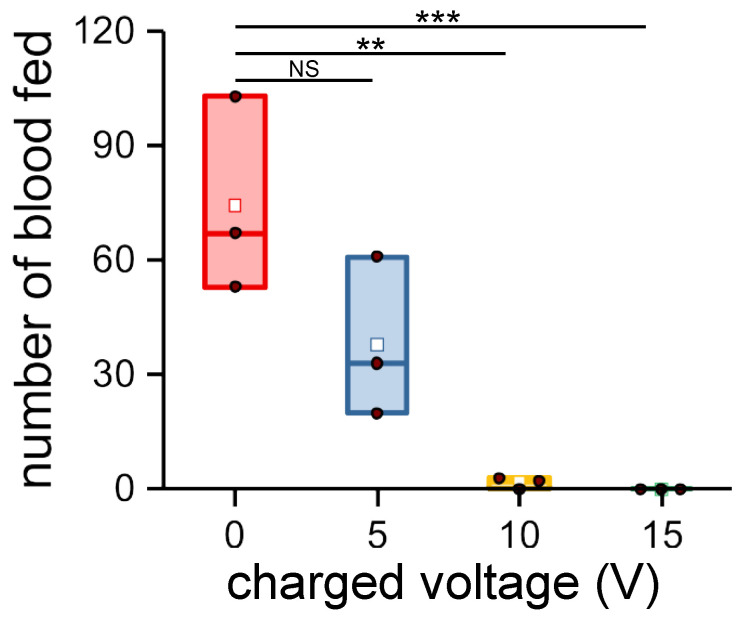
Number of mosquitoes that acquired a blood meal at different levels of charge on the MRC. The solid brown dot in the box charts represents the raw data. The height of the box depends on the 25th and 75th percentiles. The whiskers represent the 5th and 95th percentiles. Additional values including the median (line inside of the box) and mean (white dot) are also presented in the box chart. Data were analyzed by Student’s *t*-test; NS, not significant, ** *p* < 0.001, *** *p* < 0.0001.

**Figure 7 insects-14-00405-f007:**
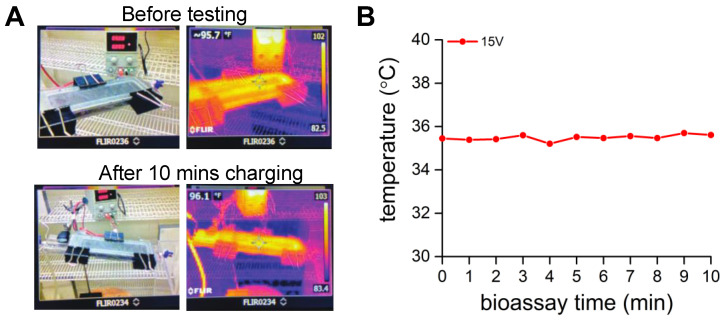
Temperature of the MRC surface. (**A**) Left is a picture of the MRC on the feeding membrane in regular light while the picture on the right was obtained using an infrared camera before (0 V) and after 10 min at 15 V; and (**B**) temperature of the MRC at each min after the application of 15 V.

**Figure 8 insects-14-00405-f008:**
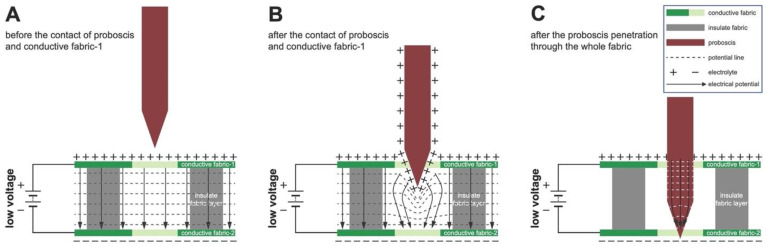
Electrical potential during mosquito probing through the MRC. (**A**) Before proboscis interacts with the MRC; (**B**) proboscis as it approaches the top and bottom conductive layers; and (**C**) where current is flowing through the proboscis (see text for details).

## Data Availability

Data available by request with justification from the corresponding author.

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
