# Peer review of "Mosquito Blood Feeding Prevention Using an Extra-Low DC Voltage Charged Cloth"

_insects, 2023, doi:10.3390/insects14050405_

Round 1

Reviewer 1 Report

Major comments

There is increasing interest in developing new ways to prevent mosquito bites given the limitations of current insecticides and repellents. The authors of this study have developed a new system for preventing mosquito blood feeding by developing a fabric with a low voltage current which is applied only when the proboscis of a mosquito pierces through the fabric. This material has the potential to reduce mosquito bites if worn as a garment.

This technology is quite exciting and the study is likely to be of broad interest with potentially significant applications. Unfortunately, the main conclusions of the study appear to be overstated as the experiments are quite limited. Firstly, the authors state that the material is 100% bite proof but this is only measured through scoring mosquito blood feeding. The experiment clearly shows that the mosquito-bite resistant cloth (MRC) prevents blood feeding at 15 volts, but it is unclear if biting is prevented since probing and biting attempts are not quantified and mosquitoes can still pierce the material with their proboscis. Secondly, the authors do no testing on humans and instead use a membrane feeding device in a small cage of 100 mosquitoes which may not provide a realistic measure of biting reduction. Thirdly, the authors provide no evidence that the material is safe as stated in the abstract.

In the discussion the authors acknowledge that this is only a proof of concept, and more research is required before it would be feasible and cost-effective to apply this to a worn garment, but I would like to see some comparisons to other more low-tech approaches. Apart from insecticide-treated fabrics, what else is out there that can reduce biting and how does this compare? For instance, what advantage will this provide over a garment that is loose-fitting (which greatly reduces the ability of mosquitoes to feed) or that has a thickness greater than the proboscis length?

I am concerned, based on the current data, as to whether the bite prevention will hold up under more realistic conditions. If the voltage is triggered only when a proboscis closes the circuit, will it be reliably triggered when only one or two mosquitoes bite? I would argue that the following experiments are required before the authors can claim that the device prevents mosquito bites: 1. the running the experiment with single mosquitoes, 2. Testing the device on a live human and 3. Measuring biting directly rather than engorgement. Otherwise, the authors must tone down their claims and state that the device prevents blood feeding on a membrane feeder rather than preventing biting.

Minor comments

Line 120 – rephrase- penetration is not necessarily easier with a membrane compared to human skin depending on the material used.

Line 159 – Provide more details about the animal blood if possible, including the species, supplier, storage conditions and if any anticoagulant was used.

Line 173- Was it at least 1 second or at least 2 seconds?

Line 189 – “whispers” should be “whiskers”

Line 220- Show the exact P value

Line 236 – The membrane feeder here is warmer than the typical surface of human skin so this experiment is not particularly informative in my opinion. What temperature does the MRC reach on its own without the heated membrane feeder?

Line 246 – Why is a cloth thickness of <2.2 mm a requirement or even desirable? Surely a cloth thicker than this would help to reduce mosquito bites regardless of any voltage applied

Figure 5 – If mosquitoes are allowed to feed and there is no distinction between fed and unfed mosquitoes then counting landings is not very informative. It is plausible that the increase when a voltage is present is due to the prevention of blood feeding but there is no way to tell.

Author Response

see file chosen.

Reviewer 2 Report

It is an exciting manuscript dealing with innovative tech approaches to stop mosquito vector bites.  The authors present results of using low voltage cloths or experimentally designed membranes to repel mosquito females feeding. Authors also describe their work as a Proof of Principle research.  Here I made some comments that may help to create a scientific frame and to understand the core idea of the manuscript better:

1. Risk assessment has to be addressed somewhere in the manuscript.  Although the voltage is low (-15 v) compared to pest control insect killers (+1500 v), the repellent device is planned to protect humans. The World Health Organization, Food and Drug Administration, and related agencies seriously consider the topic of magnetic fields. Suspects of the health impact of cell phones, microwaves, and electrical home appliances are a public concern in many countries. 

2. Is it possible to explain why the NC University Ethical Committees did not consider some risk recommendations for the project? Does the University think similarly to WHO and FDA about magnetic fields?

3. Cost-benefit always needs to be in the discussion section.

4. Which characteristics had all the blood-feeding successful mosquitos in the treatments? Is a shorter proboscis enough to penetrate the MRC without closing the circuit? Or a smaller head size? Are all this group of mosquitoes smaller in size? Does it justify making mesh adjustments?

5. Did the authors see any delayed effect on mosquito females after being repealed by the 10 and 15 volts? For instance, 24-h mortality, egg-laying reduction, and other sublethal dose effects. Can you add a short comment?

6. I suggest shortening the paragraphs about Power consumption in the discussion section unless it supports the main results obtained. 

7. Figures are excellent for understanding the experiments.  

Author Response

See file chosen.

Round 2

Reviewer 1 Report

All important issues have been addressed